# Anticancer Plant Secondary Metabolites Evicting Linker Histone H1.2 from Chromatin Activate Type I Interferon Signaling

**DOI:** 10.3390/ijms26010375

**Published:** 2025-01-04

**Authors:** Olga Vlasova, Irina Antonova, Khamis Magomedova, Alena Osipova, Polina Shtompel, Anna Borunova, Tatiana Zabotina, Gennady Belitsky, Irina Budunova, Albert Jordan, Kirill Kirsanov, Marianna Yakubovskaya

**Affiliations:** 1N. N. Blokhin National Medical Research Center of Oncology, Ministry of Health of Russia, 24 Kashirskoe Shosse, 115522 Moscow, Russiah_magomedova2000@mail.ru (K.M.);; 2SBHI Moscow Clinical Scientific Center Named After Loginov MHD, 111123 Moscow, Russia; 3Department of Dermatology, Northwestern University, Chicago, IL 60611, USA; 4Institut de Biologia Molecular de Barcelona (IBMB-CSIC), 08028 Barcelona, Spain; 5Institute of Medicine, Peoples’ Friendship University of Russia (RUDN University), Miklukho-Maklaya St. 6, 117198 Moscow, Russia

**Keywords:** plant secondary metabolites, plant polyphenols, phytochemicals, DNA-binding compounds, anticancer effects, chromatin structure, linker histone eviction, type I interferon signaling, *LINE1* transcription, double-stranded DNA ends

## Abstract

Previously we discovered that among 15 DNA-binding plant secondary metabolites (PSMs) possessing anticancer activity, 11 compounds cause depletion of the chromatin-bound linker histones H1.2 and/or H1.4. Chromatin remodeling or multiH1 knocking-down is known to promote the upregulation of repetitive elements, ultimately triggering an interferon (IFN) response. Herein, using HeLa cells and applying fluorescent reporter assay with flow cytometry, immunofluorescence staining and quantitative RT-PCR, we studied effects of PSMs both evicting linker histones from chromatin and not influencing their location in nucleus. We found that (1) 8 PSMs, evicting linker histone H1.2 from chromatin, activated significantly the type I IFN signaling pathway and out of these compounds resveratrol, berberine, genistein, delphinidin, naringenin and curcumin also caused *LINE1* expression. Fisetin and quercetin, which also induced linker histone H1.2 eviction from chromatin, significantly activated only type I IFN signaling, but not *LINE1* expression; (2) curcumin, sanguinarine and kaempferol, causing significant depletion of the chromatin-bound linker histone H1.4 but not significantly influencing H1.2 presence in chromatin, activate type I IFN signaling less intensively without any changes in *LINE1* expression; (3) four PSMs, which did not cause linker histone eviction, displayed neither IFN signaling activation nor enhancement of *LINE1* expression. Thus, we have shown for the first time that chromatin destabilization observed by depletion of chromatin-bound linker histone H1.2 caused by anticancer DNA-binding PSMs is accompanied by enhancement of type I IFN signaling, and that *LINE1* expression often impacts this activation.

## 1. Introduction

A whole number of plant secondary metabolites (PSMs) or phytochemicals, largely consisting of polyphenols, have been shown to possess anticancer activity against chemically induced animal tumors of various types [1,2,3,4]. They reduce the incidence and multiplicity of benign and malignant tumors induced in rodents in colon by 1,2-dimethylhydrazine or azoxymethane, in breast and ovary by 7,12-dimethylbenz(a)anthracene and in breast by N-methyl-N-nitrosourea. Mechanistic data, obtained on human cancer cells growing ex vivo and cultured in vitro, confirmed antiproliferative, proapoptotic, anti-inflammatory and immunomodulatory effects of PSMs [5,6,7,8]. Chemopreventive effects of genistein, resveratrol, berberine and some other PSMs were shown in a number of clinical trials [2,9,10,11].

Anticancer PSMs, characterized by the presence of aromatic rings with hydroxyl and other substituents, can bind various molecular targets in cells, such as receptors, enzymes of xenobiotic metabolism and epigenetic regulation of transcription, components of signaling pathways and enzymes of DNA repair and metabolism [12,13,14,15]. A wide range of PSM targets makes it extremely difficult to analyze consequences of different peculiar interactions, which impact the integral result of PSM action. Having unique structure, every PSM is characterized by its own spectrum of targets; however, most anticancer PSMs possess affinity to DNA. PSMs interact with DNA via van der Waals, ionic, and hydrogen bonds without forming covalent bonds, which explains why they are not genotoxic. Intercalation into DNA helix was shown for apigenin, delphinidin, fisetin, epigallocatechin-3-gallate (EGCG), genistein, naringenin, quercetin, resveratrol and sanguinarine [16,17,18,19,20,21]. Curcumin and sanguinarine interact with DNA as minor groove binders [16,22]. G-quadruplex binding and stabilization were shown for berberine, curcumin, EGCG, fisetin, kaempferol, quercetin, sanguinarine [23,24,25,26,27,28,29]. Formation of DNA-PSM complexes can affect the spatial characteristics of a DNA duplex, its flexibility and physicochemical properties, as well as its ability to form various alternative DNA structures [19,30]. PSMs can cover up DNA sites, which are recognized by enzymes of DNA repair, packaging, epigenetic regulation, transcription and replication analogously to minor grove ligands preventing interaction of Poly (ADP-ribose) Polymerase 1 with DNA duplex [31]. 

Previously we demonstrated that most of DNA-binding PSMs (11 compounds out of analyzed 15 PSMs with anticancer activity) cause eviction of linker histones H1.2 and H1.4 from chromatin [32]. Noteworthy, in multiH1 knocked-down cells, chromatin opening promotes the upregulation of repetitive elements, ultimately triggering an interferon (IFN) response [33]. In particular, Izquierdo-Bouldstridge et al. demonstrated that histones H1.2 and/or H1.4 are involved in the expression control of transposable elements (TEs). In general, H1 linker histones are enriched in the constitutive heterochromatin with silent repetitive elements *LINEs*, *SINEs*, and repeats containing endogenous retroviruses [34,35]. Chromatin-related effects of DNA-binding small molecules started to be investigated about 10 years ago, demonstrating histone eviction from chromatin by anticancer agents from the anthracycline group and a new anticancer drug Curaxin CBL0137 [36,37]. Then anticancer activity of Curaxin CBL0137 was shown to be decreased in mice with the knocked out *Interferon Alpha and Beta Receptor Subunit 1*, responsible for type I IFN signaling activation [38]. Curaxin CBL0137 anticancer activity was also reduced in severe combined immune deficient mice when compared to immune competent mice [39]. Curaxin CBL0137 ability to induce type I IFN signaling was explained by enhanced transcription of repetitive heterochromatin elements as double-stranded RNA induce this signaling pathway.

Compounds that interact with DNA without causing DNA alterations, but induce changes in chromatin structures, make constitutive heterochromatin accessible to the transcriptional machinery. The divergent transcription of centromeric and pericentromeric repeats leads to the accumulation of double-stranded RNAs. These transcripts are recognized by cytoplasmic nucleic acid sensitive receptors and activate the IFN response [38]. It should be also noted that chromatin remodeling, caused by ATP-dependent helicase, was also demonstrated to activate type I IFN signaling [40].

The activation of this signaling pathway is realized by IFNs, a broad class of cytokines, representing key modulators of the immune response. These cytokines with potent antiviral and growth-inhibitory effects play critical roles in the first line of defense against infections and homeostatic disorders during cancer pathogenesis [41,42]. IFN signaling activation was described in several studies devoted to effects of some PSMs. In particular, interferon activation was observed when cells were treated with resveratrol [43,44], fisetin [45], naringenin [46,47], sanguinarine [48], quercetin [45,49] and berberine [50,51]. All these studies were performed using single PSMs and different cancer cell lines which makes it difficult to compare their effects, and they do not show possible mechanisms of IFN activation. However, these data and our previously obtained results concerning PSM influence on linker histone location in cell nuclei provide a good basis for clarifying the question of whether PSM-induced chromatin destabilization is accompanied by IFN activation. This clarification should both expand our understanding of molecular effects induced by anticancer PSMs and reveal cell response to chromatin destabilization caused by different DNA-binding small molecules. The latter may serve as the basis for the development of new non-genotoxic chemopreventive and anticancer drugs targeting chromatin structure and function [30]. 

Thus, we propose that PSMs bind DNA and cause some distortions of the helix, which is followed by both linker histone eviction from chromatin and type I IFN signaling activation. As linker histone eviction from chromatin induces the transcription of silent repetitive elements, it may impact type I IFN signaling activation. The aims of the present study include analyzing the influence of 15 anticancer DNA-binding PSMs on IFN-signaling activity, on the patterns of IFN-responsive genes, and on the transcription of repetitive non-coding DNA. Finally, the main goal of our study was to compare the data obtained with the previously described abilities of PSMs to cause linker histones H1.2 and H1.4 evictions from chromatin [32]. We chose HeLa and T47D cells as the object of our study as previously it was on these cells that we observed linker histones H1.2 and H1.4 evictions from chromatin under PSM treatment.

## 2. Results

### 2.1. Type I Interferon Signaling Activation by DNA-Binding PSMs

In our study, we employed two alternative approaches for assessment of IFN signaling activity in HeLa cells. Firstly, we used flow cytometry and the reporter assay that revealed the IFN response through the activation of a consensus ISRE driving mCherry red fluorescent protein transgene expression in HeLa TI ISRE-mCherry cells. Previously we demonstrated that this approach of IFN response assessment is highly sensitive. Secondly, we used the Human Signal Transduction Pathway Finder RT2Profiler PCR Array (HSTPF, Qiagen, PARN-014Z, Hilden, Germany) to analyze the changes in the expression pattern of 84 INF-responsive genes. Dose-dependence for PSM toxic effects in HeLa cells was described in our previous publications, and based on those data we chose non-toxic and IC20 (leaving more than 80% of cells alive) concentrations of PSMs for our study (Appendix A). Untreated cells and cells treated with the solvents were used as negative controls, while cells treated with IFN-α were used as the positive control. As the main goal of our study was to compare PSM effects on IFN activation and *LINE1* expression with their ability to cause linker histones H1.2 and H1.4 evictions from chromatin described previously [32], the PSM order for the effect presentation in all the figures was as follows: 1–8—PSMs causing intensive linker histones eviction from chromatin (mainly H1.2, but accompanied with H1.4), 9–11—PSMs causing significant H1.4 eviction from chromatin, but insignificant depletion of chromatin-bound H1.2, and 12–15—PSMs unable to cause both H1.2 and H1.4 eviction from chromatin.

#### 2.1.1. PSM Influence on Reporter mCherry Expression Driven by IFN-Sensitive Response Element

Using the reporter assays and flow cytometry, we observed a very intensive IFN response in almost all HeLa TI ISRE-mCherry cells after 24 h treatment with berberine, curcumin, fisetin, naringenin and resveratrol, while IFN-α treatment activated mCherry expression in 99.0% of cells (Figure 1). Significantly increased proportions of the cells expressing mCherry were also observed after cell treatment with 4 PSMs, in particular, for genistein (by 70.5%) for sanguinarine (by 60.3%), for quercetin (by 33.5%) and for delphinidin (by 22.5%). We did not observe significant increases in the proportions of cells expressing mCherry after the treatment with apigenin, coumarin, ginsenoside Rb1, thymoquinone, EGCG and kaempferol. Significant increases of mCherry mean fluorescence intensity (MFI) were observed in HeLa TI ISRE-mCherry cells treated with 5 PSMs, although they were less intensive compared to the MFI in cells treated with IFN. In particular, IFNα caused MFI to increase by 13.3 times, while naringenin caused MFI to increase by 12.7 times, fisetin—by 11.8 times, curcumin and resveratrol—by 9.3 times, and berberine—by 8.9 times (Figure 1B).

For PSMs, which caused a significant increase in the proportion of cells expressing mCherry after 24 h treatment, we also analyzed the dynamics of the changes after 1, 6 and 24 h PSM treatment (Figure 2).

We observed significant increases in both the proportion of cells expressing mCherry and the mean fluorescence intensity after 1 h treatment with fisetin and resveratrol, and the effects increased in a time-dependent manner after 6 h and 24 h treatment (Figure 2). Curcumin and naringenin caused significant effects after 6 h treatment and their effects were further enhanced after 24 h treatment. Berberine, genistein, sanguinarine and delphinidin caused significant effects only after 24 h treatment. MFI increases after the treatment with curcumin, berberine and naringenin were observed only after 24 h treatment.

Thus, using reporter assay and flow cytometry, we demonstrated the activation of the type I IFN signaling pathway after treating HeLa TI ISRE-mCherry cells for 24 h with 9 PSMs. Two of them (resveratrol and fisetin) caused a significant increase in INF activation even after 1 h treatment.

#### 2.1.2. Influence of PSMs on the Expression Pattern of IFN-Responsive Genes

The expression pattern of IFN-responsive genes was analyzed after 24 h treatment with 15 PSMs (at non-toxic concentrations) and with IFN-α (10^3^ U/mL) as the positive control and using the Human Signal Transduction Pathway Finder RT2Profiler PCR Array (HSTPF, Qiagen, PARN-014Z) (Figure 3).

HeLa cells 24 h treatment with IFN-α, which was used as a positive control, caused a significant expression increase of all the IFN-mediated genes analyzed. Analogous cell treatment with a number of PSMs, in particular, fisetin, curcumin, berberine, resveratrol, genistein, quercetin, kaempferol, naringenin and delphinidin, caused expression activation of the most genes in the type I IFN signaling pathway (Figure 3, Appendix A). However, significant increases were observed only for a number of individual genes: for quercetin—9 genes, for resveratrol and fisetin—7 genes, for curcumin and naringenin—4 genes, for berberine and genistein—2 genes, and for thymoquinone—1 gene. Noteworthy, in the treated cells among the 84 genes of the type I IFN signaling pathway the number of genes whose expression levels were more than doubled was 62 for resveratrol, 61 for quercetin, 59 for fisetin, 54 for berberine, 53 for naringenin, 52 for curcumin, 48 for genistein, 49 for kaempferol, and 46 for delphinidin. It is also worth noting that 14 genes turned out to be common for the following compounds: *CXCL10*, *IFI27*, *IFNA14*, *IFNA16*, *IFNA2*, *IFNA4*, *IFNA7*, *IFNG*, *IL20RA*, *IL5RA*, *IRF8*, *IRGM*, and *ISG15*. We observed that a larger number of type I IFN signaling pathway genes enhanced their expression when cells were treated with sanguinarine, kaempferol, and EGCG; however, these effects were not significant. We observed very low activation levels or even weak inactivation for the genes of the type I IFN signaling pathway following apigenin, coumarin, ginsenoside Rb1 and thymoquinone treatment.

Thus, using the Human Signal Transduction Pathway Finder RT2Profiler PCR Array, we also observed activation of the type I IFN signaling pathway after HeLa cell treatment with 8 of the 15 studied PSMs (numbered from 1 to 8, as they cause eviction of the linker histones H1.2 and H1.4), and 4 PSMs (numbered from 12 to 15, as they do not cause eviction of the linker histones H1.2 and H1.4) did not produce unidirectional influence (Appendix A).

We studied the effect of resveratrol, genistein, apigenin and thymoquinone at non- toxic concentrations on the expression activation of type I IFN signaling genes (IFN-responsive genes *IFI27* and *OASL*, and IFN regulatory factor *IRF1*) in another cell line, T47D (human breast cancer cells). According to our previous data, resveratrol and genistein cause active depletion of chromatin-bound linker histones H1.2 and H1.4 both in HeLa and in T47D cells, while apigenin and thymoquinone did not demonstrate this effect on the HeLa cells. We observed an increase in relative gene expression, which was significant for gene *IRF1* in both cell lines after the treatment of cells with resveratrol and genistein, while the levels of relative expression were comparable to those of the controls after the treatment with apigenin or thymoquinone. The data obtained are comparable between two cell lines, indicating a general pattern of observations (Figure 3B–D).

### 2.2. Induction of LINE1 Expression by Some PSMs

Influence of PSMs on *LINE1* expression was studied using two alternative methods: (1) qRT-PCR assessment of the expression levels of three *LINE1* amplicons (A, B, C) and the encoded in *LINE1* gene *ORF1 LINE1* of nucleic acid-binding protein, which is essential for retrotransposition of LINE-1, and (2) immunofluorescence/flow cytometry analysis of the cells with stained *ORF1 LINE1* and γ-H2AX proteins [52].

#### 2.2.1. Quantitative Estimation of *LINE1* Expression Level in HeLa Cells Treated with PSMs

The most pronounced activation of the expression of transposable *LINE1* sequences was observed when HeLa cells were treated with delphinidin. This PSM treatment caused the expression levels of *LINE1* amplicons A, B and C to enhance by 3.7; 3.8 and 3.6 times, respectively, and also caused the expression level of the *ORF1 LINE1* gene to increase by 4.9 times (Figure 4).

For resveratrol, naringenin, genistein and berberine, the average expression of *LINE1* amplicons increased by 1.7; 2.0; 2.5 and 1.9 times, and the expression of *ORF1 LINE1* increased by 1.9; 2.2; 2.2 and 2.1 times, respectively. Cell treatment with curcumin was followed by a significant increase in the expression level of *ORF1 LINE1* by 2.2 times. The described changes in the expression levels of *LINE1* amplicons and the *ORF1 LINE1* gene were significant. Treatment of cells with other PSMs used in the study did not cause a statistically significant increase in the expression levels of *LINE1* amplicons and/or *ORF1 LINE1* gene. Thus, in this part of the study we demonstrated that 6 PSMs, in particular curcumin, berberine, delphinidin, naringenin, genistein and resveratrol, can enhance the activity of *LINE1* expression.

#### 2.2.2. Analysis of the Amount of *ORF1 LINE1* and γ-H2AX Proteins by Flow Cytometry in HeLa Cells Treated with PSM

For analysis of *LINE1* activity we used immunofluorescent staining with antibodies to *ORF1 LINE1* and γ-H2AX proteins and analyzed populations of treated and untreated cells using flow cytometry. *LINE1* retrotransposition causes the appearance of DNA double-stranded breaks, and therefore γ-H2AX was used as one of the markers of active retrotransposition, despite the fact that it is not specific to the target gene. However, it is considered to be a marker of the retrotransposition process. We also controlled the proportion of apoptotic cells (lower than 5%) to prevent possible interference of apoptosis and *LINE1* retrotransposition (Figure 5A).

The most pronounced effect was observed for delphinidin. After 24 h treatment with delphinidin the proportion of the cells expressing *LINE1 ORF1* protein increased by 2.9 times, and the average fluorescence intensity associated with γ-H2AX appearance increased by 3.8 times. Noteworthy, with an exposure time of 72 h, delphinidin did not induce significant changes. Genistein, on the contrary, caused a statistically significant increase in the average fluorescence intensity associated with both *ORF1 LINE1* and γ-H2AX at an exposure time of 24 h by 2.0 times and 2.8 times, respectively, and after 72 h treatment at a concentration of 60 μM it caused the increase by 3.6 and 8.5 times, respectively. For fisetin, only an increase in parameters associated with γ-H2AX was observed at an exposure time of 24 h. Resveratrol caused the statistically significant increase of the average fluorescence intensity associated with *ORF1 LINE1* (3.7 times).

Thus, using an alternative, if less sensitive approach, we confirmed that some PSMs affect the *LINE1* expression level.

## 3. Discussion

Over the last thirty years, there has been great progress in understanding innate and adaptive immunity thanks to the discovery of different pattern recognition receptors (PRRs), which were shown to be associated with pathogens [53,54,55].

Four major sub-families of PRRs comprise more than 400 of receptors, in particular toll-like receptors (TLR), nucleotide-binding oligomerization domain–Leucin–Rich Repeats-containing receptors (NLR), retinoic acid-inducible gene 1 (RIG-1)-like receptors, and the C-type lectin receptors [53]. Recent PRR discoveries allow us to present in detail the genius idea of the “chemical binding of exogenous substances to cell”, which dominated Paul Erlich’s life more than a hundred years ago [54,56]. Based on the available information about immune system functioning and the role of PRR activation in immune responses, Polly Matzinger elaborated “the Danger theory” that immune system responses are less concerned with the self/unself origin of antigens than with the context of their influence on tissue homeostasis [57,58,59,60]. When PRRs interact with their ligands, corresponding to danger-associated molecular patterns (DAMPs), they induce a cell response. This response is in turn manifested as significant changes in cell signaling, including type I IFN signaling activation, which is considered to focus on identifying viral nucleic acids in the midst of exceedingly host-derived RNA and DNA [61].

From this point of view, agents which interact with DNA and alter DNA–protein interactions should cause the appearance of DAMPs, as these changes could influence the pattern of the transcribed sequences, including both coding and non-coding DNA. Recently, we have found that a number of PSMs could cause linker histone eviction from chromatin [32]. It may thus be proposed that PRRs recognize PSM-DNA complexes or some other internal structures appearing after PSM-DNA complex formation.

We separated the studied PSMs into groups depending on their ability to cause linker histone eviction from chromatin: compounds **1** to **11** demonstrated this ability in our previous experiments, while compounds **12** to **15** did not (Figure 6) [32]. A more comprehensive analysis of the type I IFN signaling activation pattern where using the Human Signal Transduction Pathway Finder RT2Profiler PCR Array showed results that correspond perfectly to our hypothesis: we revealed IFN signaling activation under the treatment of compounds **1**–**11**. However, it should be pointed out that while compounds **9**–**11** caused significant eviction of only the linker histone H1.4, their effects on type I IFN signaling were lower than the effects caused by compounds inducing significant eviction of H1.2.

These data mainly correspond to the results of our ISRE-mCherry reporter analysis, and we are able to explain some small discrepancies between the PSM effects by the fact that in this part of the study we only used ISRE to assess IFN signaling activation. The result shows that PSMs 12–15, which did not cause linker histone eviction from chromatin, are not able to activate type I IFN signaling that was observed in both types of experiments, when two alternative techniques of IFN signaling analysis were applied.

The activation of type I IFN signaling by certain PSMs has not been studied very intensively, so published data are scarce. However, our results correspond to the published data available. In particular, it was shown that quercetin and fisetin activate *IFN-α* in RAW 264.7 cells [50]. In the study of Lin et al., resveratrol was shown to induce TLR9 activation of IFN-β signaling [45]. Activation of IFN-β signaling was also revealed in RAW264.7 and HEK293T cells after the treatment with berberine [47]. Naringenin induced *IFN-α* activation in U2OS cells, which was demonstrated both by luciferase reporter assay and RT-PCR methods [48]. Sanguinarine was shown to enhance type I INF signaling in cultured monocyte-derived macrophages [49]. In the study of Ullah et al., contradictory results were published regarding genistein ability to influence type I IFN signaling: using STING competent mouse L929 cells demonstrated genistein positive effect, stably expressing an ISRE-luciferase, while the same cells cocultured with STING-deficient cGAS-overexpressing human HEK-cGASlow cells showed the opposite effect, caused by STING blocking [62]. At the same time, another study reported the antiviral activity of genistein, which is considered to be the result of type I IFN signaling activation [63].

Concerning PSMs 12–15 which do not cause chromatin-bound linker histone depletion, the following data were published: thymoquinone was shown to actually decrease type I IFN signaling activity in RAW 264.7 and MCF-7 cells [64]; for ginsenoside RB1 in CRFK 157 cells after 48 h treatment, no activation of type I IFN signaling was observed [65]; apigenin was shown to influence the inhibitory effect of IFN-α on cancer cell viability, wherein said viability is mediated by 26S proteasome inhibition (however, the effect of apigenin itself on type I IFN signaling was not analyzed) [66].

Our research on the activation of IFN signaling by PSMs also revealed that PSMs 3–8, which caused significant depletion of the chromatin-bound linker histone H1.2 and H1.4, activate *LINE1* expression. This observation concurs with the results of Izquierdo-Bouldstridge et al. who demonstrated that histones H1.2 and/or H1.4 participate in repression of repeats [33]. It also perfectly corresponds to a well-known fact that *LINE1* expression along with other TEs stimulate type I IFN signaling [34,67]. PSMs 9–11, which caused more significant depletion of the only chromatin-bound linker histone H1.4, induce IFN signaling less actively. A proposed explanation is that this is the consequence of the differential presence of H1 variants within transposable element classes and families described by Salinas-Pena et al. [34]. For instance, in T47D cells and to some extent also in HeLa cells, H1.2 and H1.4 are enriched in different TEs, meaning that H1.4 is enriched in evolutionarily recent SVA, Alu, L1 and LTR, while H1.2 is enriched in older TEs. Noteworthy, fisetin and quercetin (PSMs 1 and 2 in our study) did not induce *LINE1* expression, although they did cause significant depletion of the histones H1.2 and H1.4. It is also worth mentioning that their chemical structures are very similar (Appendix A). Thus, we revealed that the influence of PSMs on chromatin structure via linker histone eviction from chromatin may be accompanied by *LINE1* expression enhancement, which in turn impacts type I IFN signaling activation and, consequently, impacts the anticancer activity of the corresponding PSMs.

Additionally, a literature review showed that specific direct or indirect influence on PRR-induced signaling had already been described for the PSMs studied. In particular, it has already been shown that fisetin binds to TLR4, inhibits the binding of lipopolysaccharide (LPS) to the TLR4/MD2 complex and attenuates inflammatory reaction via the TLR4/NLRP3 inflammasome pathway [46,68,69,70,71], while quercetin and resveratrol inhibit TLR4 and inflammasome activation [72,73,74,75]. Genistein [76] and berberine [45] were also demonstrated to have anti-inflammatory effects via suppression of the TLR4-mediated signaling pathway. Naringenin suppresses inflammatory responses by regulation of cell-surface TLR2 functioning [47]. Delphinidin inhibits LPS-induced TLR4, MUC8 and MUC5B expression [77]. Curcumin was reported to inhibit extracellular TLR 2 and 4 and intracellular TLR9 [78]. Kaempferol attenuates TLR4/NF-κB pathway activation in LPS-activated BV2 cells [79]. Sanguinarine inhibits the TLR4/NF-κB pathway in H9c2 cardiomyocytes and thus attenuates LPS-induced inflammation [80], while it up-regulates expressions of endosomal TLRs [49]. EGCG was also revealed to suppress LPS-induced TLR4 activity [81,82]. Apigenin inhibits the LPS-mediated inflammatory mediator production in keratinocytes by reducing the TLR4-dependent activation of Akt, mTOR and NF-κB pathways [83,84]. Thymoquinone was shown to block the TLR4/NF-κB signaling pathway in microglia cells [85]. Ginsenoside Rb1 reduces TLR4 dimerization followed by inhibiting the TLR4-MyD88-NF-κB/MAPK pathways [86]. Coumarins were shown to attenuate inflammation also via TLRs [87]. Thus, anti-inflammatory effects were described for all the PSMs considered in our study, which should impact anticancer activity along with type I IFN signaling accompanying linker histone eviction from chromatin.

## 4. Materials and Methods

### 4.1. Cell Culture

The HeLa cell line was obtained from the Blokhin CRC cell collection. HeLa-TI-ISRE-mCherry cells were kindly provided by Dr. Katherina Gurova from the Department of Cell Stress Biology at the Roswell Park Cancer Center (Buffalo, NY, USA). Preparation and maintenance of HeLa-TI-ISRE-mCherry cells, containing integrated red fluorescent protein (mCherry) gene, driven by a consensus IFN-sensitive response element (ISRE), were described previously [38]. Human breast cancer cells T47D were kindly provided by Dr. Albert Jordan from the Department of Molecular Genomics at the Molecular Biology Institute of Barcelona of IBMB-CSIC (Barcelona, Catalonia, Spain). Cells were cultured in Dulbecco’s Modified Eagle Medium (DMEM, C420p, PanEco, Moscow, Russia) supplemented with L-glutamine (0.584 mg/mL) (F033E, PanEco, Moscow, Russia), penicillin (50 U/mL), streptomycin (50 µg/mL) (A063p, PanEco, Moscow, Russia) and 10% fetal bovine serum (Biowest, S1810-500, Nuaillé, France). Cell lines were incubated at 37 °C and 5% CO_2_. All cell lines were validated by STR profiling and tested negative for mycoplasma.

### 4.2. Plant Secondary Metabolites

All of the studied compounds were obtained from Chemlight, Moscow, Russia. We studied apigenin (CAS 520-36-5); berberine (CAS 633-65-8); coumarin (CAS 91-64-5); curcumin (CAS 458-37-7); delphinidin (CAS 13270-61-6); EGCG (CAS 989-51-5); fisetin (CAS 528-48-3); genistein (CAS 446-72-0); ginsenoside Rb1 (CAS 41753-43-9); kaempferol (CAS 520-18-3); naringenin (CAS 480-41-1); quercetin (CAS 117-39-5); resveratrol (CAS 501-36-0); sanguinarine chloride hydrate (CAS 5578-73-4) and thymoquinone (CAS 490-91-5).

### 4.3. Other Chemicals and Reagents

Curaxin CBL0137 was provided by Incuron, Inc., Moscow, Russia. TRIzol™ Reagent (15596026), Moloney Murine Leukemia Virus Reverse Transcriptase (M-MLV RT) (18057018) and Random(dN)10 (SB002) were purchased from Evrogen, Moscow, Russia. dNTP mix, dye and primers were purchased from Evrogen, Moscow, Russia. Triton X-100 (CAS 9002-93-1) was purchased from BioInnlabs, Rostov-on-Don, Russia. Dimethyl sulfoxide (DMSO, 67-68-5|102952), cOmplete™, Mini Protease Inhibitor Cocktail (cat. 11836153001), phosphate-buffered saline (PBS, P4417), bovine serum albumin (CAS 9048-46-8), IFN-α A protein, recombinant human albumin (P01563) were purchased from Sigma–Aldrich (Merck), Bengaluru, Karnataka, India. Versene Solution (P080p), Trypsin-EDTA 0.25% solution with Hanks salts (P043p) and phosphate-buffered saline (PBS, P4417) were purchased from PanEco, Moscow, Russia. DC™ Protein Assay Kit I (5000111EDU) was purchased from Bio-Rad (Moscow, Russia). Clarity Max™ Western ECL Substrate for Chemiluminescent Detection of Horseradish Peroxidase (HPR) Conjugates (cat. 1705062) was purchased from Helicon, Moscow, Russia. The 2.5× Reaction mixture for qRT-PCR in the presence of SYBR Green I dye (M-427) was purchased from Syntol (Moscow, Russia). Antibodies *LINE1*-*ORF1* (cat# MABC1152, 1:500) were purchased from Sigma–Aldrich (Merck), Bengaluru, Karnataka, India; γ-H2AX (cat# ab26350, 1:700) and Donkey Anti-Mouse IgG H&L (Alexa Fluor^®^ 488, cat# ab150105; 1:1000) were purchased from Abcam, Cambridge, UK).

### 4.4. Quantitative Reverse Transcriptase–Polymerase Chain Reaction for Analysis of Expression of LINE1 and PSM-Induced Interferon Signaling

For the assay, tumor cells (HeLa) were seeded in 6-well plates (10^5^ cells per well in 2 mL DMEM) and incubated with various concentrations of compounds for 24 h and IFN-α (10^3^ UI/mL) was used as a positive control for the IFN signaling analysis. Total RNA was then extracted using TRIzol™ Reagent according to the manufacturer protocol. cDNA was synthesized using a reverse transcription reaction. Total RNA (1 μg, from both control and treated cells) was reverse-transcribed using M-MLV RT reverse transcriptase and random Random(dN)10 in a reaction volume of 20 μL according to the manufacturer protocol (Evrogen, Russia). RNA quantification was performed using NanoDrop Lite (ThermoScientific, Waltham, MA, USA). 

For the analysis of the expression of *LINE1* amplicones, qRT-PCR was carried out in a reaction mixture containing Master Mix (0.3 mM dNTP mix (10 mM each), 3 mM MgCl_2_, nuclease-free deionized water, SYBR^®^ Green dye, 10X Taq Turbo Buffer, 0.2 U/µL Taq DNA polymerase), 0.2 µM forward and reverse primers and 5 ng of DNA template, in accordance with the manufacturer protocol (Evrogen, Russia). Thermal cycling conditions were as follows: initial denaturation step by heating at 95 °C for 5 min, followed by 40 cycles of 15 s initial denaturation (at 95 °C), 20 s at the appropriate melting temperature according to the primers, and 25 s extension at 72 °C. Expression of the gene of interest was normalized to the constitutively expressed housekeeping genes RPL0 and HAPDH. The relative expression level was calculated for each sample using the 2^−ΔΔCt^ method. All experiments were performed at least in triplicate biological replicates.

The sequences of the gene-specific primers used for qRT-PCR were as follows (Primer design from [52]):LINE1_amplA_F: 5′GCCAAGATGGCCGAATAGGA 3′LINE1_amplA_R: 5′AAATCACCCGTCTTCTGCGT 3′ LINE1_amplB_F: 5′CGAGATCAAACTGCAAGGCG 3′LINE1_amplB_R: 5′CCGGCCGCTTTGTTTACCTA 3′LINE1_amplC_F: 5′ TAAACAAAGCGGCCGGGAA 3′LINE1_amplC_R: 5′ AGAGGTGGAGCCTACAGAGG 3′LINE1_ORF1_F: 5′ ACCTGAAAGTGACGGGGAGA 3′ LINE1_ORF1_R: 5′CCTGCCTTGCTAGATTGGGG 3′RPL0 F: 5′CCTTCTCCTTTGGGCTGGTCATCC A 3′RPL0 R: 5′CAGACACTGGCAACATTGCGGACAC 3′HAPDH F: 5′GTCTCCTCTGACTTCAACAGCG 3′HAPDH R: 5′ACCACCCTGTTGCTGTAGCCAA 3′

The sequences of the gene-specific primers used for type I IFN signaling qRT-PCR were as follows (Primer design from [33]):IFI27_F: 5′ TGCTCTCACCTCATCAGCAGT 3′IFI27_R: 5′ CACAACTCCTCCAATCACAACT 3′ OASL_F: 5′ GGGACAGAGATGGCACTGAT 3′OASL_R: 5′ AAATGCTCCTGCCTCAGAAA 3′ IRF1_F: 5′ TTTGTATCGGCCTGTGTGAATG 3′IRF1_R: 5′ AAGCATGGCTGGGACATCA 3′ 

For analysis of gene expression of type I IFN signaling qRT-PCR was performed in 96-well Human Signal Transduction PathwayFinder™ RT 2 Profiler™ PCR Array plates (https://geneglobe.qiagen.com/us/product-groups/rt2-profiler-pcr-arrays/PAHS-064Z (accessed on 24 July 2024), Qiagen, PAHS-064Z, Hilden, Germany) according to the manufacturer protocol: 95 °C for 10 min, then 40 cycles of 95 °C for 15 s and 60 °C for 1 min. Each RT2 Profiler PCR array contains gene-specific primers for qRT-PCR assays for a carefully screened set of 84 genes, consisting of IFNs, IFN receptors, IFN regulatory factors, and IFN-responsive genes (Table 1).

Expression of genes of interest was normalized to constitutively expressed housekeeping genes (*ACTB*, *B2M*, *GAPDH*, *HPRT1*, *RPLP0*). Relative expression levels were calculated for each sample using the 2^−ΔΔCt^ method using the manufacturer software. All experiments were performed at least in triplicate biological replicates.

### 4.5. Analysis of ISRE-mCherry Reporter Activation in HeLa-TI-ISRE-mCherry Cells by Flow Cytometry

IFN response in HeLa-TI-ISRE-mCherry cells treated with PSMs was assessed by the proportion of the cells expressing mCherry driven by ISRE as well as by mCherry mean fluorescence intensity (MFI) using a BD FACSCanto™ II flow cytometer (BD Biosciences, San Jose, CA, USA). Cells were seeded in 6-well plates (10^5^ cells per well in 2 mL DMEM) and incubated with PSM at non-toxic concentrations for 24 h. For PSMs inducing IFN response after 24 h treatment we studied the dynamics of their effects at 1, 6 and 24 h. After the treatment with PSMs cells were removed from the culture plates using Versene Solution and 0.25% trypsin-EDTA and washed with PBS. To maintain high cell viability, a PBS solution with 2% fetal bovine serum was used as a cell storage buffer. The concentration of dimethyl sulfoxide (DMSO) in the medium for all compounds did not exceed 0.01%. All experiments were performed in triplicate biological replicates. The obtained data were analyzed using WinList™ 3D software (Version 9.0.1, Verity Software House, https://www.vsh.com/products/winlist/index.asp, accessed on 10 September 2024, Topsham, ME, USA).

### 4.6. Analysis of PSM Induced LINE1 Activation by Immunofluorescent Antibody Staining and Flow Cytometry 

To analyze PSM induced *LINE1* activation, HeLa cells were seeded in 6-well plates (10^5^ cells per well in 2 mL DMEM). After 24 h, cells were treated with compounds of interest at IC20 or non-toxic concentrations and incubated for 24/72 h. Then, the cells were removed from the substrate with trypsin, washed three times with PBS and fixed in cold 4% paraformaldehyde for 15 min. After three washes with PBS, the cells were permeabilized with cold 0.3% Triton-X100 for 7 min and blocked with bovine serum albumin for 1 h. Cells were immunofluorescently stained with antibodies to LINE1-ORF1, γ-H2AX, and subsequent binding with secondary antibodies AlexaFluor488 was carried out in the dark. Cells were washed with PBS and analyzed on a BD FACSCanto™ II flow cytometer (BD Biosciences, San Jose, CA, USA). Proportions of the cells positive for the fluorescent signal and the average intensity of the cell fluorescence normalized to the control were assessed. The obtained data were analyzed using WinList™ 3D software (Version 9.0.1, Verity Software House, https://www.vsh.com/products/winlist/index.asp, accessed on 10 September 2024, Topsham, ME, USA).

### 4.7. Annexin-FITC/Propidium Iodide Double Staining 

Cells were stained with annexin V-FITC and PI to evaluate apoptosis by flow cytometry according to the manufacturer instructions to the FITC Anexin V Apoptosis Detection Kit I (Sigma-Aldrich, St. Louis, MO, USA). Cells were treated with maximum non-toxic concentrations of PSM for 24 h. After treatment, cells were collected, washed twice with ice-cold PBS, and resuspended in 0.5 mL of annexin/V-FITC/PI solution for 30 min in the dark according to manufacturer protocol. After staining at room temperature, cells were analyzed using the BD FACSCanto™ II flow cytometer (BD Biosciences, San Jose, CA, USA). For each sample, 10,000 events were acquired and positive FITC and/or PI cells were quantified using WinList™ 3D software (Version 9.0.1, Verity Software House, https://www.vsh.com/products/winlist/index.asp, accessed on 10 September 2024, Topsham, ME, USA).

### 4.8. Statistical Analysis

We compared the data from the experimental and control groups using one-way analysis of variance (ANOVA) and Dunnett’s post hoc test. Differences between groups were considered significant at a *p*-value < 0.05. The basis for the statistical processing of results to determine the presence of statistically significant differences between several groups for one independent variable is the randomness of the samples, the equality of the sample size and the normality of the distributions of the samples used. The normality of data distribution was assessed with the Kolmogorov–Smirnov test. Statistical analyses were performed using GraphPad Prism 8.3.0 (GraphPad Software Inc., San Diego, CA, USA).

## 5. Conclusions

PSMs are important chemical components in plants and are actively used in human nutrition. The active use of PSMs may be explained by their influence on human health, an effect which is considered to be a result of coevolution of flora and fauna. Our study revealed that linker histone H1.2 eviction from chromatin, caused by a number of natural DNA-binding anticancer small molecules known as PSMs, is accompanied by their activating influence on type I IFN signaling. In contrast, PSMs, not influencing linker histone H1.2 locations in nucleus, do not change type I IFN signaling activity. The findings of this study allow us to propose a new mechanism for type I IFN signaling activation via environmental DNA-binding small molecules presented by PSMs, which cause chromatin destabilization. This is in agreement with “the Danger theory” proposed by Polly Matzinger; however, it requires additional studies to elucidate damage-associated molecular patterns formed after PSM-DNA interaction, as well as their PRRs and peculiar gene targets of activated PRRs that represent a new field of PSM investigations. Moreover, further studies of various PSMs with rather similar as well as different chemical structures in comparison with modeling data of their binding to different DNA motives and their effects on the locations of various linker histones will reveal PSM structural peculiarities related to activating the type I IFN response, which enables anticancer immunity. Also, in further studies, it would be interesting to expand the range of model cell lines of various histogeneses and to assess the contribution of interferon signaling activation by the studied compounds to the antitumor effect in vivo. These data are important for the elaboration of a new type anticancer drugs, which are chromatin destabilizers activating antitumor immunity.

## Figures and Tables

**Figure 1 ijms-26-00375-f001:**
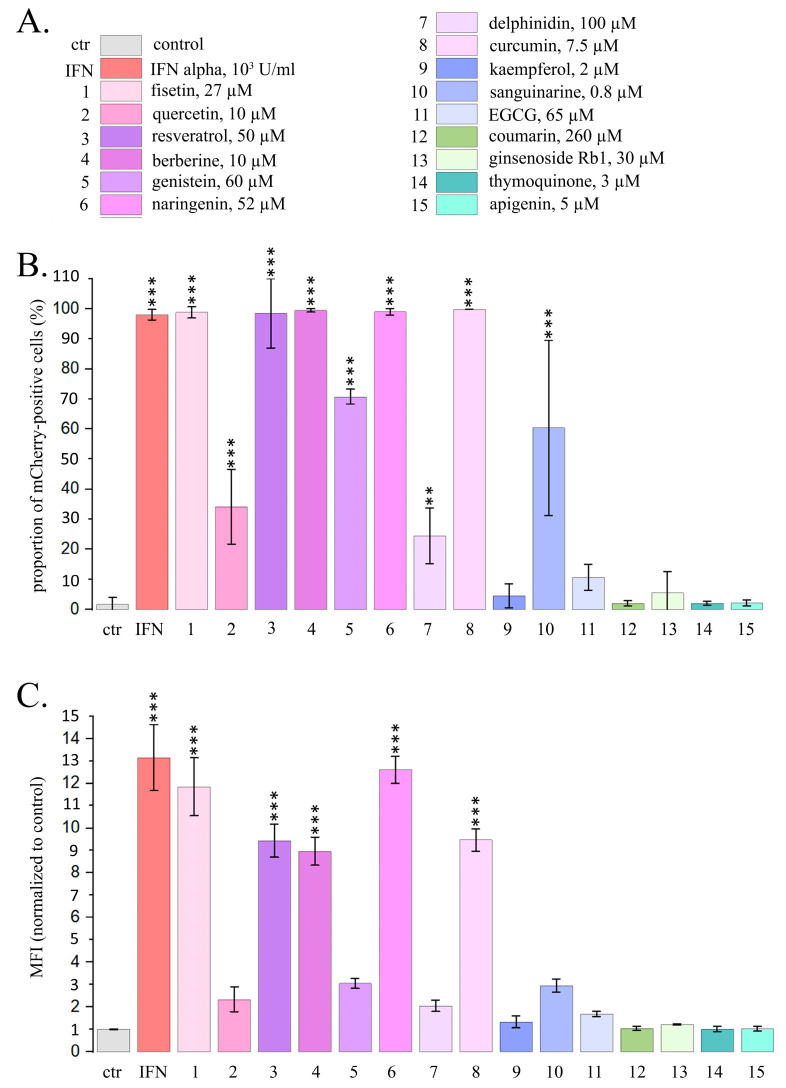
Flow cytometry data for the expression of mCherry driven by IFN-sensitive responsive element (ISRE) in HeLa TI ISRE-mCherry cells after PSM treatment for 24 h. (**A**) Color–numeric designation of PSMs and their non-toxic concentrations. Ctr—control; IFN—IFN-α, 10^3^ U/mL; 1—fisetin, 27 µM; 2—quercetin, 10 µM; 3—resveratrol, 50 µM; 4—berberine, 10 µM; 5—genistein, 60 µM; 6—naringenin, 52 µM; 7—delphinidin, 100 µM; 8—curcumin, 7.5 µM; 9—kaempferol, 2 µM; 10—sanguinarine, 0.8 µM; 11—EGCG, 65 µM; 12—coumarin, 260 µM; 13—ginsenoside Rb1, 30 µM; 14—thymoquinone, 3 µM; 15—apigenin, 5 µM. This color-number legend is used in all figures. (**B**) Proportions of the cells expressing mCherry. (**C**) Mean fluorescence intensity of mCherry per cell (normalized to control). The data are presented as an average value ± SD. Significance of the differences between control untreated cells and PSM-treated cells was determined using ANOVA test and Dunnett’s post hoc test: significant difference, **—*p* < 0.01, ***—*p* < 0.001.

**Figure 2 ijms-26-00375-f002:**
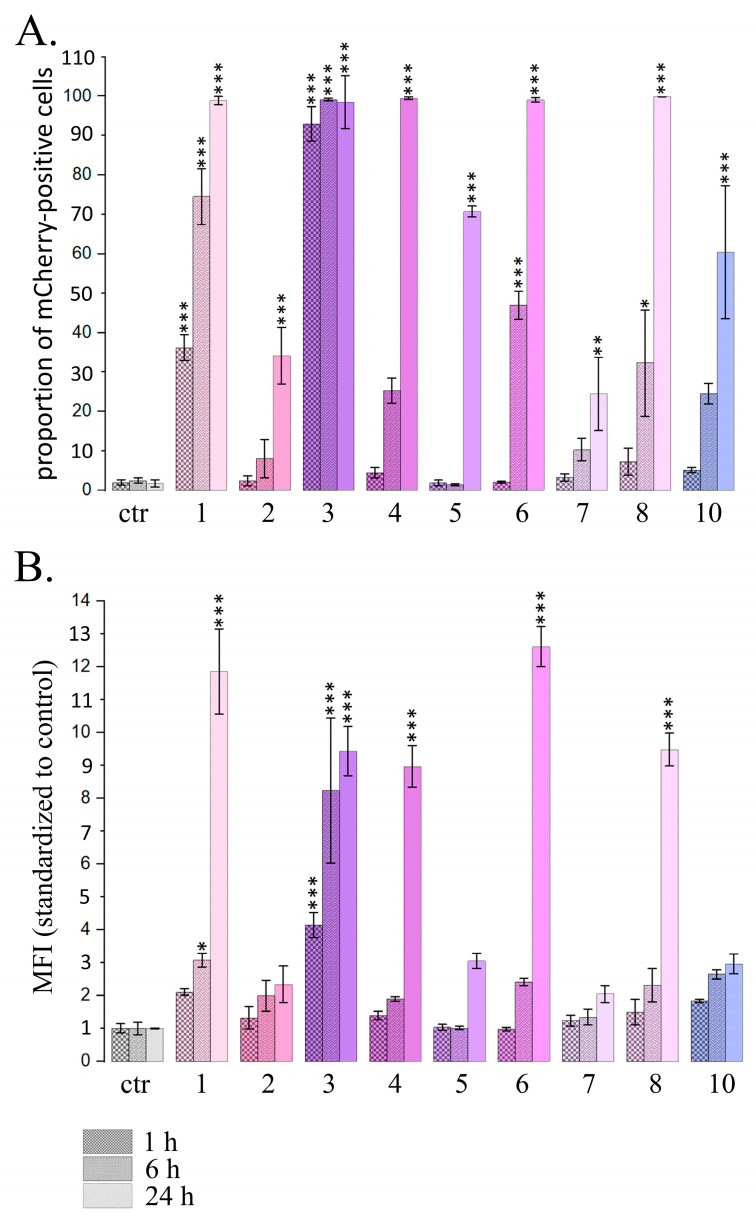
Flow cytometry data for the expression of mCherry driven by IFN-sensitive responsive element (ISRE) in HeLa TI ISRE-mCherry cells after PSMs treatment for 1, 6 and 24 h. Ctr—control; 1—fisetin, 27 µM; 2—quercetin, 10 µM; 3—resveratrol, 50 µM; 4—berberine, 10 µM; 5—genistein, 60 µM; 6—naringenin, 52 µM; 7—delphinidin, 100 µM; 8—curcumin, 7.5 µM; 10—sanguinarine, 0.8 µM. (**A**) Proportions of the cells expressing mCherry. (**B**) Mean fluorescence intensity of mCherry per cell (normalized to control). The data are presented as M ± SD. Significance of the differences between control untreated cells and PSM treated cells was determined using ANOVA test and Dunnett’s post hoc test: significant difference, *—*p* < 0.05, **—*p* < 0.01, ***—*p* < 0.001.

**Figure 3 ijms-26-00375-f003:**
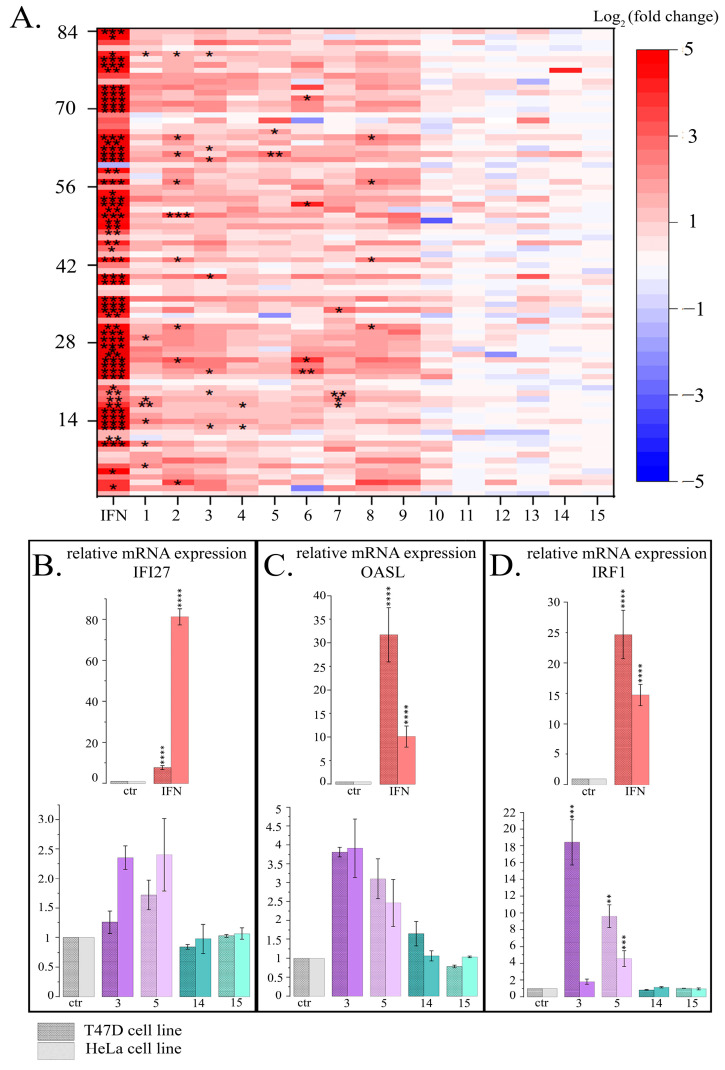
Influence of PSMs on gene expression of type I IFN signaling pathway. (**A**) Pattern of gene expression of type I IFN signaling. Data presented as Log_2_ (fold change) for each of the 84 genes (HSTPF, Qiagen, PARN-014Z). IFN- IFN-α, 10^3^ U/mL; 1—fisetin, 27 µM; 2—quercetin, 10 µM; 3—resveratrol, 50 µM; 4—berberine, 10 µM; 5—genistein, 60 µM; 6—naringenin, 52 µM; 7—delphinidin, 100 µM; 8—curcumin, 7.5 µM; 9—kaempferol, 2 µM; 10—sanguinarine, 0.8 µM; 11—EGCG, 65 µM; 12—coumarin, 260 µM; 13—ginsenoside Rb1, 30 µM; 14—thymoquinone, 3 µM; 15—apigenin, 5 µM. Significance of the differences between control untreated cells and PSM treated cells was determined using ANOVA test and Dunnett’s post hoc test: significant difference, *—*p* < 0.05, **—*p* < 0.01, ***—*p* < 0.001. (**B**–**D**). mRNA expression of genes IFN-signaling normalized to *RPL0* and *HAPDH* in T47D and HeLa cell lines. Ctr—control; IFN—IFN-α, 10^3^ U/mL; 3—resveratrol, 50 µM; 5—genistein, 60 µM; 14—thymoquinone, 3 µM; 15—apigenin, 5 µM. The data are presented as M ± SD. Significance of the differences between control untreated cells and PSM treated cells was determined using ANOVA test and Dunnett’s post hoc test: significant difference, **—*p* < 0.01, ***—*p* < 0.001, ****—*p* < 0.0001. (**B**) IFN-responsive gene *IFI27*. (**C**) IFN-responsive gene *OASL*. (**D**) IFN regulatory factor *IRF1*.

**Figure 4 ijms-26-00375-f004:**
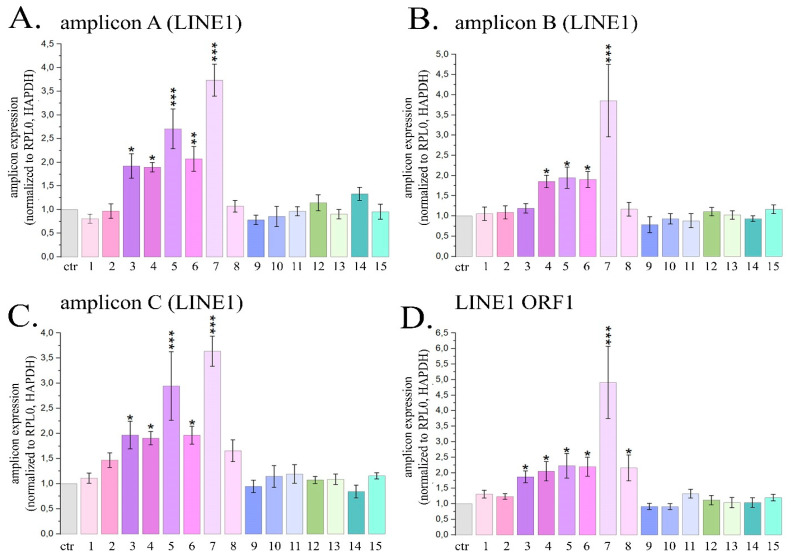
Expression of three *LINE1* amplicons (**A**–**C**) and *ORF1 LINE1* gene (**D**) in HeLa cells treated with maximal non-toxic concentrations of PSMs for 24 h. Ctr—control; 1—fisetin, 27 µM; 2—quercetin, 10 µM; 3—resveratrol, 50 µM; 4—berberine, 10 µM; 5—genistein, 60 µM; 6—naringenin, 52 µM; 7 –delphinidin, 100 µM; 8—curcumin, 7.5 µM; 9—kaempferol, 2 µM; 10—sanguinarine, 0.8 µM; 11—EGCG, 65 µM; 12—coumarin, 260 µM; 13—ginsenoside Rb1, 30 µM; 14—thymoquinone, 3 µM; 15—apigenin, 5 µM. The data are presented as M ± SD. Significance of the differences between PSM treated cells and control untreated cells was determined using ANOVA test and Dunnett’s post hoc test: significant difference, *—*p* < 0.05, **—*p* < 0.01, ***—*p* < 0.001.

**Figure 5 ijms-26-00375-f005:**
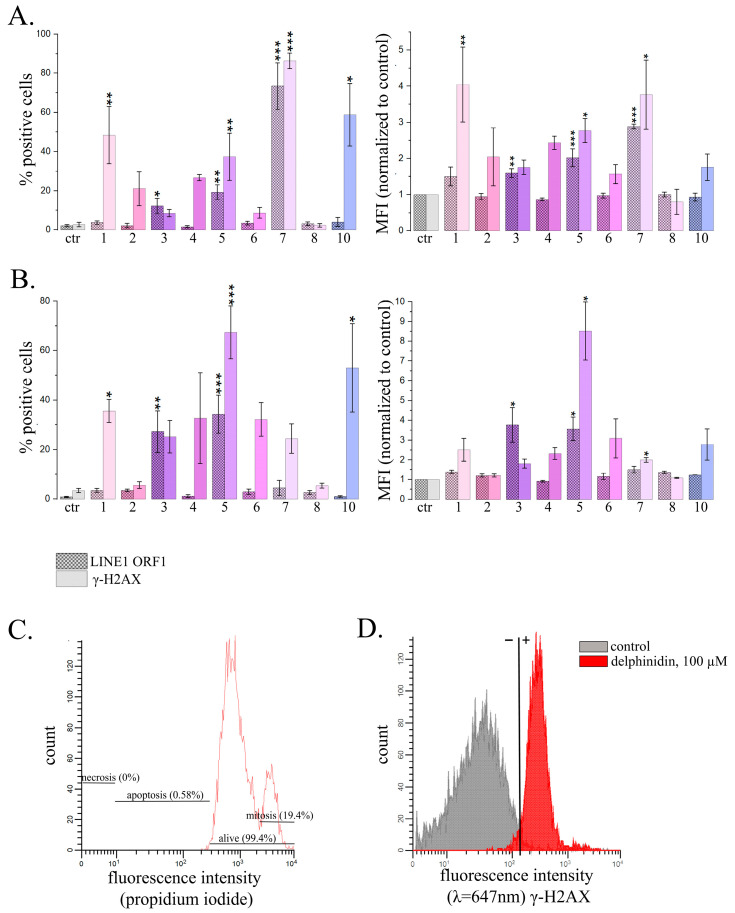
Flow cytometry analysis of HeLa cells treated with PSMs in non-toxic concentrations and immunofluorescently stained *ORF1 LINE1* or γ-H2AX. (**A**,**B**) The effects of 24 h treatment with PSMs on the proportions of stained cells. Ctr—control; 1—fisetin, 27 µM; 2—quercetin, 10 µM; 3—resveratrol, 50 µM; 4—berberine, 10 µM; 5—genistein, 60 µM; 6—naringenin, 52 µM; 7—delphinidin, 100 µM; 8—curcumin, 7.5 µM; 10—sanguinarine, 0.8 µM. (**B**) PSM treatment for 72 h. Ctr—control; 1—fisetin, 13.5 µM; 2—quercetin, 5 µM; 3—resveratrol, 25 µM; 4—berberine, 5 µM; 5—genistein, 30 µM; 6—naringenin, 26 µM; 7—delphinidin, 50 µM; 8—curcumin, 3.7 µM; 10—sanguinarine, 0.4 µM. The data are presented as M ± SD. Significance of the differences between control untreated cells and PSM treated cells was determined using ANOVA test and Dunnett’s post hoc test: significant difference, *—*p* < 0.05, **—*p* < 0.01, ***—*p* < 0.001. (**C**) Measurement of proportion of apoptotic cells in the analyzed populations of fixed cells. (**D**) HeLa cells treated with delphinidin for 24 h with the immunofluorescently stained γ-H2AX.

**Figure 6 ijms-26-00375-f006:**
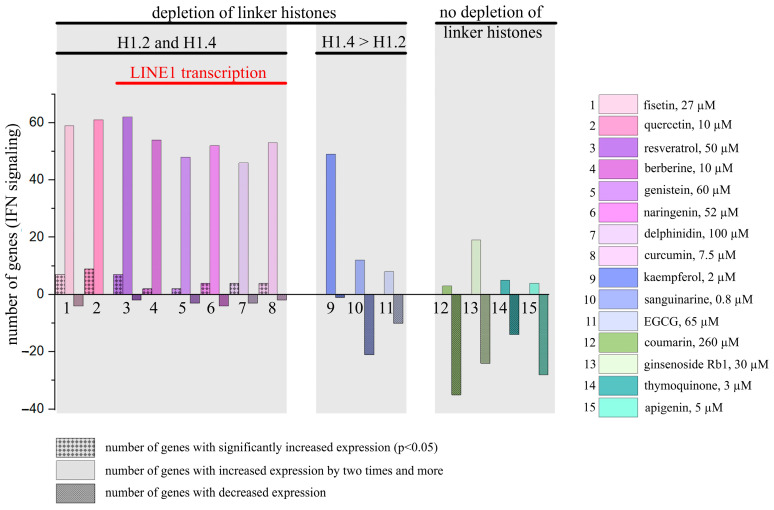
PSM influence on the type I IFN signaling genes. PSMs were separated into groups depending on their ability to cause eviction of linker histones from chromatin.

**Table 1 ijms-26-00375-t001:** Description of the set of 84 genes analysed.

Type of Gene Products	Gene Products	Number of Genes
IFNs (21)	IFN-α; IFN-β; receptor ligands	5 genes
	IFN-γ; receptor ligands	1 genes
	Hematopoietin & IFN class (D200-domain) cytokine receptor ligands	10 genes
	Other IFN related genes	5
IFN receptors (37)	IFN-α and IFN-β receptors	2
	IFN-γ receptors	2
	Hematopoietin, IFN class (D200-domain) receptors	28
IFN regulatory factors (9)		9
IFN-responsive genes (23)	Response to virus	13 *
	Transcriptional regulation	2 *
	Other IFN responsive genes	8

* Note: The *IFI16* gene is repeated in groups of Response to virus and transcriptional regulation.

## Data Availability

Data presented in this study are contained within this article and in the Appendix A, or are available upon request to the corresponding author.

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
