# Peer review of "Anticancer Plant Secondary Metabolites Evicting Linker Histone H1.2 from Chromatin Activate Type I Interferon Signaling"

_ijms, 2025, doi:10.3390/ijms26010375_

Round 1
Reviewer 1 Report
Comments and Suggestions for Authors
International Journal of Molecular Sciences (Manuscript ID: ijms-3293243), Comments to the Authors:
Title: Anticancer Plant Secondary Metabolites Modulating Chromatin Activate Interferon Signaling and LINE1 Expression
Comments
The submitted manuscript highlighted the use of HeLa cells and applyed fluorescent reporter assay with flow cytometry and quantitative RT-PCR, to study PSMs depleting linker histones. They found 8 compounds significantly activate type I Interferon (IFN) signaling pathway. Resveratrol, berberine, genistein, delphinidin, naringenin and curcumin caused LINE1 expression detected by quantitate RT-PCR and by immunofluorescent staining analysis of ORF1 LINE1 and γ-H2AX. Fisetin and quercetin, which also induced linker histone depletion, significantly activated only type I IFN signaling, but not LINE1 expression. Curcumin, sanguinarine and kaempferol, causing depletion of the linker histone H1.4 more intensively respectively to H1.2, activate IFN signaling less intensively without any changes of LINE1 expression.
I think the submitted paper can be accepted after the authors respond to the following comments:
1. The title is not clear, the title should be rephrased. The authors should be more specific with their language especially with the words “Anticancer Plant Secondary Metabolites”.
2. The abstract is disorganized and should be rewritten. The authors should provide numerical values of the activity to show the readers the significance of their work.
3. Some sentences are complex and could be simplified for better readability. For example, the sentence “When PRRs interact with their ligands, presented as the exogenous molecules (pathogen and environmental chemicals) as well as some internal structures associated with homeostasis disorders, it induces cell response” could be broken down into shorter sentences.
4. The conclusion should be rewritten to give a more focused summary of the work.
5. The authors should indicate the future applications of their work on the development of anticancer agents from natural products.
6
Comments on the Quality of English Language. There are many typos and spelling mistakes that should be corrected before resubmission.
Author Response
To: Peer Reviewer 1 (manuscript ID – JIMS-3293243)
Dear Peer Reviewer, we are very grateful for your careful reading of our manuscript, rational comments and valuable pieces of advice, as well as for appreciation of our study. Please, find the description of corrections or rebuttal of each point that was raised in your review.
Comment 1: The title is not clear, the title should be rephrased. The authors should be more specific with their language especially with the words “Anticancer Plant Secondary Metabolites”.
Response for the comment 1: We have rephrased the title so that it corresponds more to our discoveries. Our study is devoted to DNA-mediated effects caused by plant secondary metabolites, which possess anticancer activity and interact with DNA while not forming covalent bonds. In our previous publication (10.31083/j.fbl2908275) we used the term “Anticancer plant secondary metabolites” as chemicals, phytochemicals and polyphenols, which harbor anticancer activity or prevent chemically induced or spontaneous cancers, are called, respectively, “anticancer chemicals” (doi: 10.3390/biomedicines12010201; 10.3389/fphar.2024.1387866; 10.3892/ol.2013.1679), “anticancer phytochemicals” (doi: 10.3389/fpls.2016.01667; 10.1002/fsn3.4318; 10.3390/molecules28176251) and “anticancer polyphenols” (10.1155/2016/6475624; 10.1021/acsapm.2c01316; 10.2174/1574892817666220512220036). All four reviewers of our previous publication and two other reviewers of this our manuscript did not mind us using this term. We would like to ask you to let us use the term “Anticancer plant secondary metabolites” in the title of this work.
Previous title: Anticancer Plant Secondary Metabolites Modulating Chromatin Activate Interferon Signaling and LINE1 Expression
New version of the title: Anticancer Plant Secondary Metabolites Evicting Linker Histone H1.2 from chromatin Activate Type I Interferon Signaling
Comment 2: The abstract is disorganized and should be rewritten. The authors should provide numerical values of the activity to show the readers the significance of their work.
Response for the comment 2: We have reorganized the abstract following your advice:
New version of the Abstract:
Previously we discovered that among 15 DNA-binding plant secondary metabolites (PSMs) possessing anticancer activity, 11 compounds cause depletion of the chromatin-bound linker histones H1.2 and/or H1.4. Chromatin remodeling or multiH1 knocking-down is known to promote the upregulation of repetitive elements, ultimately triggering an interferon response. Herein, using HeLa cells and applying fluorescent reporter assay with flow cytometry, immunofluorescence staining and quantitative RT-PCR, we studied effects of PSMs both evicting linker histones from chromatin and not influencing their location in nucleus. We found that (1) 8 PSMs, evicting linker histone H1.2 from chromatin, activated significantly type I Interferon (IFN) signaling pathway and out of these compounds resveratrol, berberine, genistein, delphinidin, naringenin and curcumin also caused LINE1 expression. Fisetin and quercetin, which also induced linker histone H1.2 eviction from chromatin, significantly activated only type I IFN signaling, but not LINE1 expression; (2) curcumin, sanguinarine and kaempferol, causing significant depletion of the chromatin-bound linker histone H1.4 but not significantly influencing H1.2 presence in chromatin, activate type I IFN signaling less intensively without any changes of LINE1 expression; (3) four PSMs, which did not cause linker histone eviction, displayed neither IFN signaling activation nor enhancement of LINE1 expression. Thus, we have shown for the first time that chromatin destabilization observed by depletion of chromatin-bound linker histone H1.2 caused by anticancer DNA-binding PSMs is accompanied by enhancement of type I IFN signaling, and LINE1 expression often impacts this activation.
Comment 3: Some sentences are complex and could be simplified for better readability. For example, the sentence “When PRRs interact with their ligands, presented as the exogenous molecules (pathogen and environmental chemicals) as well as some internal structures associated with homeostasis disorders, it induces cell response” could be broken down into shorter sentences.
Response for the comment 3: We have tried to simplify complex sentences.
In particular, we removed the details about the nature of the ligands from the sentence in comment 3. Such details make the sentence too heavy, and though originally we found this information interesting, it is not required here, since DAMPs are not the object of our study.
Previous version: When PRRs interact with their ligands, presented as the exogenous molecules (pathogen and environmental chemicals) as well as some internal structures associated with homeostasis disorders, it induces cell response.
New version: When PRRs interact with their ligands, corresponding to danger-associated molecular pattern (DAMPs), it induces cell response.
Comment 4: The conclusion should be rewritten to give a more focused summary of the work.
Response for the comment 3: We have rewritten our Conclusion following your advice:
Previous version of the Conclusions: PSMs are important chemical components of the plants, actively used in human nutrition. Their active use may be explained by PSMs influence on human health and considered to be a result of coevolution of flora and fauna. Out study revealed that linker histone depletion from chromatin fraction, caused by DNA-binding anticancer PSMs, is linked to their activating influence on type I IFN signaling. It let us propose a new mechanism of type I IFN signaling activation by environmental chemicals, which cause chromatin remodeling. It is in agreement with “the Danger theory” proposed by P.Matzinger, however it requires additional studies to elucidate damage associated molecular patterns formed after PSM-DNA interaction, as well as their PRRs and peculiar gene targets of activated PRRs, that represent a new field of PSM investigations. Moreover, further studies are needed to compare the effects of PSMs with rather similar as well as different chemical structures to examine peculiar new data, important to develop anticancer chromatin remodeling drugs.
New version of the Conclusions: PSMs are important chemical components of the plants, actively used in human nutrition. Their active use may be explained by PSMs influence on human health and is considered to be a result of coevolution of flora and fauna. Our study revealed that linker histone H1.2 eviction from chromatin, caused by a number of DNA-binding anticancer natural small molecules known as PSMs, is accompanied by their activating influence on type I IFN signaling. In contrast, PSMs, not influencing linker histone H1.2 locations in nucleus, do not change type I IFN signaling activity. It let us propose a new mechanism of type I IFN signaling activation by environmental DNA-binding small molecules presented by PSMs, which cause chromatin destabilization. It is in agreement with “the Danger theory” proposed by P. Matzinger, however, it requires additional studies to elucidate damage associated molecular patterns formed after PSM-DNA interaction, as well as their PRRs and peculiar gene targets of activated PRRs that represent a new field of PSM investigations. Moreover, further studies of various PSMs with rather similar as well as different chemical structures in comparison with modeling data of their binding to different DNA motives and their effects on the locations of various linker histones will reveal PSM structural peculiarities activating type I IFN response, which enables anticancer immunity. Also, in further studies, it would be interesting to expand the range of model cell lines of various histogenesis and to assess the contribution of interferon signaling activation by the studied com-pounds to the antitumor effect in vivo. These data are important for elaboration of anticancer drugs of a new type, which are chromatin destabilizers activating antitumor immunity.
Comment 5: The authors should indicate the future applications of their work on the development of anticancer agents from natural products.
Response for the comment 5: We have changed our Conclusion to include future applications (response to comment 4).
Comments on the Quality of English Language
There are many typos and spelling mistakes that should be corrected before resubmission.
Response: We are awfully sorry for the typos. There was a technical error during submission. We did not check if our final file transferred well and if it replaced the first file that we had uploaded when we got acquainted with the submission procedure. The error also explains the confusing situation with the different titles in the IJMS system and in the manuscript. The revised version has been edited by a professional translator.
Once again, thank you for very valuable comments.
Sincerely yours,
Olga Vlasova, MSc, junior scientist
of the Department of chemical carcinogenesis,
Institute of Carcinogenesis
Blokhin National Medical Research Center of Oncology
+79256761167
Marianna Yakubovskaya, MD, PhD, DSc,
Head of the Department of chemical carcinogenesis,
Institute of Carcinogenesis
Blokhin National Medical Research Center of Oncology
+79256761167

Reviewer 2 Report
Comments and Suggestions for Authors
The text below contains comments on manuscript entitles “Interferon Signaling Activation and LINE1 Expression under Influence of Anticancer Plant Secondary Metabolites”.
The manuscript is focused to study the influence of 15 anticarcinogenic DNA-binding on INF-signaling activity, on the patterns of INF-responsive genes and on transcription of repetitive non-coding DNA. In addition, were compared the data obtained with the abilities of PSMs to cause linker histones H1.2 and H1.4 depletions.
I encourage the authors to make a substantial revision of the English language and grammar. Other suggestions for corrections are listed here:
To my opinion, the whole abstract should be reworded. In this version, it is hardly to understand it many certain points.
Page 1 in the Abstract. It is quantitative RT-PCR, not quontitave.
Page 2: PSMs can cover up DNA sites, which are recognized by enzymes of DNA packaging, epigenetic regulation, repair, transcription and replication. – A reference is missing.
Page 2: IFNs represent key modulators of the immune response. – Do not start the sentence with an abbreviation. On the same paragraph you need a reference at the end.
The last paragraph from the introduction is more suitable to be moved to results section. If it is a part of the aim of the study it should be reworded.
Page 13, Section 4.2. Plant secondary metabolite. Please give more arguments for the selection of these compounds to be included in the experiments. Probably these arguments are better to be included in the introduction or result section.
Page 13, Section 4.4. I would advise the authors to think if the primer design and Table 1 can be placed as a supplementary.
Comments on the Quality of English LanguageMajor improvement of the quality of English Language is necessary.
Author Response
To: Peer Reviewer 2 (manuscript ID – JIMS-3293243) 25.11.2024
Dear Peer Reviewer, we are very grateful for your careful reading of our manuscript, rational comments and valuable pieces of advice. Please, find the description of corrections or rebuttal of each point that was raised in your review.
Comment 1: To my opinion, the whole abstract should be reworded. In this version, it is hardly to understand it many certain points.
Response to comment 1: The abstract has been rewritten, following the advice of the first reviewer, who proposes to provide numerical values of the facts found.
New version of the Abstract:
Previously we discovered that among 15 DNA-binding plant secondary metabolites (PSMs) possessing anticancer activity, 11 compounds cause depletion of the chromatin-bound linker histones H1.2 and/or H1.4. Chromatin remodeling or multiH1 knocking-down is known to promote the upregulation of repetitive elements, ultimately triggering an interferon response. Herein, using HeLa cells and applying fluorescent reporter assay with flow cytometry, immunofluorescence staining and quantitative RT-PCR, we studied effects of PSMs both evicting linker histones from chromatin and not influencing their location in nucleus. We found that (1) 8 PSMs, evicting linker histone H1.2 from chromatin, activated significantly type I Interferon (IFN) signaling pathway and out of these compounds resveratrol, berberine, genistein, delphinidin, naringenin and curcumin also caused LINE1 expression. Fisetin and quercetin, which also induced linker histone H1.2 eviction from chromatin, significantly activated only type I IFN signaling, but not LINE1 expression; (2) curcumin, sanguinarine and kaempferol, causing significant depletion of the chromatin-bound linker histone H1.4 but not significantly influencing H1.2 presence in chromatin, activate type I IFN signaling less intensively without any changes of LINE1 expression; (3) four PSMs, which did not cause linker histone eviction, displayed neither IFN signaling activation nor enhancement of LINE1 expression. Thus, we have shown for the first time that chromatin destabilization observed by depletion of chromatin-bound linker histone H1.2 caused by anticancer DNA-binding PSMs is accompanied by enhancement of type I IFN signaling, and LINE1 expression often impacts this activation.
Comment 2: Page 1 in the Abstract. It is quantitative RT-PCR, not quontitave.
Response to the comment 2: We are awfully sorry for the typos. There was a technical error during submission. We did not check if our final file transferred well and if it replaced the first file that we had uploaded when we got acquainted with the submission procedure. The error also explains the confusing situation with the different titles in the IJMS system and in the manuscript.
Comment 3: Page 2: PSMs can cover up DNA sites, which are recognized by enzymes of DNA packaging, epigenetic regulation, repair, transcription and replication. – A reference is missing.
Response to the comment 3: We added two references – the first one is devoted to effects of small DNA-binding molecules destabilizing chromatin structure and the second one – DNA-mediated inhibiting effect of minor grove ligands on PARP1 activation.
Formation of DNA-PSM complexes can affect the spatial characteristics of DNA du-plex, its flexibility and physicochemical properties, as well as its ability to form various alternative DNA structures [19, Luzhin et al., doi: 10.1093/nar/gkad865]. PSMs can cover up DNA sites, which are recognized by enzymes of DNA repair, packaging, epigenetic regulation, transcription and replication analogously to minor grove ligands preventing interaction of PARP1 with DNA duplex [Kirsanov doi: 10.18632/oncotarget.1742].
Comment 4: Page 2: IFNs represent key modulators of the immune response. – Do not start the sentence with an abbreviation. On the same paragraph you need a reference at the end.
Response to the comment 4: We have rewritten the first sentence and added the reference at the end of the paragraph.
Activation of this signaling pathway is realized by IFNs, a broad class of cytokines, representing key modulators of the immune response. These cytokines with potent antiviral and growth-inhibitory effects play critical roles in the first line of defense against infections and homeostatic disorders during cancer pathogenesis [36; 37]. IFN signaling activation was described in several studies devoted to effects of some PSMs. In particular, IFN activation was observed when cells were treated with resveratrol [38, 39], berberine [40, 37], fisetin [42], naringenin [43, 44], sanguinarine [45], quercetin [46]. All these studies were performed using single PSMs and different cancer cell lines which makes it difficult to compare their effects, and they do not show possible mechanisms of IFN activation. However, these data and our previously obtained results concerning PSM influence on linker histone location in cell nuclei provide a good basis for clarifying the question of whether PSM-induced chromatin destabilization is accompanied by IFN activation. This clarification should both expand our understanding of molecular effects induced by anticancer PSMs and reveal cell response on chromatin destabilization caused by different DNA-binding small molecules. The latter may serve as the basis for the development of new non-genotoxic chemopreventive and anticancer drugs targeting chromatin structure and function [Luzhin et al., doi: 10.1093/nar/gkad865].
Comment 5: The last paragraph from the introduction is more suitable to be moved to results section. If it is a part of the aim of the study it should be reworded.
Response to the comment 5: We have rewritten the last paragraph of the introduction. We have also added our hypothesis according to the advice of Peer Reviewer 2.
Thus, we propose that PSMs binds DNA and cause some distortions of the helix. It is followed both by linker histone eviction from chromatin and type I IFN signaling activation. As linker histone eviction from chromatin induce transcription of silent repetitive elements, it may impact type I IFN signaling activation. The aims of the present study include analyzing the infuence of 15 anticancer DNA-binding PSMs on IFN-signaling activity, on the patterns of IFN-responsive genes and on transcription of repetitive non-coding DNA. At last, the main goal of our study was to compare the data obtained with the previously described abilities of PSMs to cause linker histones H1.2 and H1.4 evictions from chromatin [30]. We chose HeLa and T47D cells as the object of our study as previously it was on these cells that we observed linker histones H1.2 and H1.4 evictions from chromatin under PSM treatment.
Comment 6: Page 13, Section 4.2. Plant secondary metabolite. Please give more arguments for the selection of these compounds to be included in the experiments. Probably these arguments are better to be included in the introduction or result section.
Response to the comment 6:
We have tried to explain our choice of the PSMs in the Introduction where we describe the following of their properties: (1) anticancer activity, (2) ability to bind DNA, (3) various abilities to destabilize chromatin found in our previous work.
Comment 7: Page 13, Section 4.4. I would advise the authors to think if the primer design and Table 1 can be placed as a supplementary.
Response to the comment 7:
Table 1 shows groups of genes considered in our study. We believe it to be important information for our readers. If you insist, we will move it to supplementary.
As for the primer sequences, we find it useful to compare the primer sequences used in our work with the ones presented in publications. We believe it will be inconvenient for some readers if we move it to supplementary.
Comments on the Quality of English Language
Major improvement of the quality of English Language is necessary.
Response to the comment: The revised version has been edited by a professional translator.
Once again, thank you for useful comments.
Sincerely yours,
Olga Vlasova, MSc, junior scientist
of the Department of chemical carcinogenesis,
Institute of Carcinogenesis
Blokhin National Medical Research Center of Oncology
+79256761167
Marianna Yakubovskaya, MD, PhD, DSc,
Head of the Department of chemical carcinogenesis,
Institute of Carcinogenesis
Blokhin National Medical Research Center of Oncology
+79256761167

Reviewer 3 Report
Comments and Suggestions for Authors
The authors present a strong background on plant secondary metabolites (PSMs) and their anticancer effects. Nonetheless, the justification for investigating the relationship among LINE1 expression, interferon signaling, and PSM-induced chromatin remodeling could be reinforced. I suggest elaborating on why this link is essential to investigate and how it contributes to understanding cancer biology.
A brief explanation of the relationship between interferon signaling and cancer could improve the introduction and better orient readers to the relevance of this study.
Some technical terms, such as "ORF1 LINE1," are used without explanation. Brief definitions of these terms would be helpful for non-expert readers.
To effectively frame the study objectives, consider adding a clear statement of the study's hypotheses at the end of the introduction section.
The authors have thoroughly studied the impact of PSMs on LINE1 expression and interferon signaling. However, the study's applicability is limited by the use of only HeLa cells. I recommend you add more cell lines—especially non-cancerous ones—which would improve the findings and increase their applicability.
The PSM concentrations that are employed differ widely. A standardized approach to dose selection would improve comparability across compounds.
The time points chosen (1h, 6h, 24h) may miss important early or late effects. I recommend you consider the time course, particularly for compounds showing delayed effects, which could provide a more comprehensive view of temporal changes.
Readers may find it easier to understand the study design and key points if a schematic diagram illustrating the proposed mechanism of PSM effect on interferon signalling and LINE1 expression is included. It is just a suggestion.
Extensive information on how different PSMs activate interferon signalling is included in this section. A brief description of the most important findings at the start or end of this section could make it easier to read and assist readers in identifying the primary conclusions.
The rationale for the order of PSM presentation, particularly regarding histone depletion, could be introduced earlier in the results section to clarify the sequencing.
In Figure 1, please consider a consistent y-axis scale across panels B and C to facilitate comparison between different PSMs.
The time-course data in Figure 2 provides valuable information. Additional interpretation of the different temporal patterns observed among various PSMs would further enhance the analysis. The y-axis labels are cut off in the figure 2. Please adjust the figure layout to ensure all labels are visible.
The results from the RT-PCR array are presented, yet the biological significance of gene expression changes is not fully explored. The analysis would be improved by classifying the affected genes into functional groups and discussing their consequences.
The authors do not provide enough detail on the statistical analysis methods used. Including this information, along with justifications for chosen tests, is necessary.
Certain figures, like Figure 1, may be overly complex. Splitting into multiple figures or using alternative visualization methods might improve interpretation.
Clearly explain the results in light of the body of knowledge already available on PSMs, chromatin remodeling, and interferon signaling.
Add recent studies from 2020-2024 to ensure the discussion reflects the latest advancements in the field. Address any limitations of the study and suggest avenues for future research.
Ensure all abbreviations are defined upon first use, and consider providing a list of abbreviations for reader convenience.
A thorough proofreading is recommended to improve the clarity and flow of the manuscript.
The authors link histone depletion, interferon signaling, and LINE1 expression, causal relationships are not definitively established. Additional experiments or a more cautious interpretation would be helpful to improve the quality of the manuscript. Consider performing additional statistical analyses, especially for gene expression data, to further support the conclusions.
Author Response
To: Peer Reviewer 3 (manuscript ID – JIMS-3293243) 25.11.2024
Dear Peer Reviewer,
We are very much grateful for your careful reading our manuscript, your huge work done to improve our manuscript, your rational comments and valuable pieces of advice. Please, find the description of corrections or rebuttal of each point that was raised in your review.
Comment 1. The authors present a strong background on plant secondary metabolites (PSMs) and their anticancer effects. Nonetheless, the justification for investigating the relationship among LINE1 expression, interferon signaling, and PSM-induced chromatin remodeling could be reinforced. I suggest elaborating on why this link is essential to investigate and how it contributes to understanding cancer biology.
Response to the comment 1. Thank you very much for this comment. We added corresponding information to the Introduction.
Chromatin-related effects of DNA-binding small molecules started to be investigated about 10 years ago, demonstrating of histone eviction from chromatin by anticancer agents from anthracycline group and anticancer drug Curaxin CBL0137 [Pang et al.,doi: 10.1038/ncomms2921; Safina et al.,doi: 10.1093/nar/gkw1366]. Then anticancer activity of Curaxin CBL0137 was shown to be decreased in mice with knocked out IFNAR1, responsible for type I IFN signaling activation, and it was reduced in immune deficient SCID mice when compared to immune competent mice [34, 33439292]. Curaxin CBL0137 ability to induce type I IFN signaling was explained by enhanced transcription of repetitive heterochromatin elements as double-stranded RNA induce this signaling pathway.
Compounds that interact with DNA without causing DNA alterations, but induce changes in chromatin structure make constitutive heterochromatin accessible to the transcriptional machinery. Divergent transcription of centromeric and pericentromeric repeats leads to the accumulation of double-stranded RNA. It is recognized by cytoplasmic nucleic acid sensitive receptors and activates the IFN response [29400649]. It should be also noted that chromatin remodeling, caused by ATRX protein, was also demonstrated to activate type I IFN signaling [35].
Comment 2. A brief explanation of the relationship between interferon signaling and cancer could improve the introduction and better orient readers to the relevance of this study.
Response to the comment 2. We have reorganized the introduction trying to follow your advice. In particular, we have added some information concerning non-genotoxic chromatin destabilizing compounds and type I IFN signaling activation by double-stranded RNA and following consequences.
Comment 3. Some technical terms, such as "ORF1 LINE1," are used without explanation. Brief definitions of these terms would be helpful for non-expert readers.
Response for the comment 3. We have added some explanations to the text and also made a list of abbreviations to help our readers.
We have also rewritten the text concerning ORF1 LINE1:
Influence of PSMs on LINE1 expression was studied using two alternative methods: (1) assessment by qRT-PCR of the expression levels of three LINE1 amplicons (A, B, C) and the encoded in LINE1 gene ORF1 LINE1 of nucleic acid-binding protein, which is essential for retrotransposition of LINE-1, and (2) immunofluorescence/flow cytometry analysis of the cells with stained ORF1 LINE1 and γ-H2AX proteins.
Comment 4. To effectively frame the study objectives, consider adding a clear statement of the study's hypotheses at the end of the introduction section.
Response for the comment 4: We have added study's hypotheses at the end of the introduction section:
We propose that PSMs bind DNA and cause some distortions of the helix. It is followed both by linker histone eviction from chromatin and type I IFN signaling activation. As linker histone eviction from chromatin induce transcription of silent repetitive elements, it may impact type I IFN signaling activation.
Comment 5. The authors have thoroughly studied the impact of PSMs on LINE1 expression and interferon signaling. However, the study's applicability is limited by the use of only HeLa cells. I recommend you add more cell lines—especially non-cancerous ones—which would improve the findings and increase their applicability.
Response for the comment 5. We entirely agree that applicability of our study will be extended when the phenomenon found is reproduced on the cancer and normal cells of different histogenesis in vitro and then in vivo.
However, our present study was devoted to revealing this phenomenon, for which we analyzed the effects of 15 compounds used in two or three concentration and at different times on human cervical adenocarcinoma cells HeLa.
Moreover, we used alternative technical approaches to be sure in our results. We also used breast cancer cells T47D to analyze our findings. We believe that publication of the results obtained in our rather huge study will speed up the progress in this important field of investigation of the effects of DNA-binding small molecules destabilizing chromatin.
Comment 6. The PSM concentrations that are employed differ widely. A standardized approach to dose selection would improve comparability across compounds.
Response for the comment 6. Our standardized approach is based on the fact that we chose PSM concentration analyzing their toxicity, and then we analyzed effects of the maximal non-toxic concentration. Moreover, the concentrations used in our study mainly correspond to the concentrations used by other authors of published PSM studies.
Comment 7. The time points chosen (1h, 6h, 24h) may miss important early or late effects. I recommend you consider the time course, particularly for compounds showing delayed effects, which could provide a more comprehensive view of temporal changes.
Response for the comment 7. In the beginning of the study we performed several pilot experiments and found the time points, which allow to reveal the effects. We agree that now it is possible to study further time dependence of the effects of single PSMs, however it was not the goal of the presented study devoted to illuminating new PSM properties.
Comment 8. Readers may find it easier to understand the study design and key points if a schematic diagram illustrating the proposed mechanism of PSM effect on interferon signaling and LINE1 expression is included. It is just a suggestion.
Response for the comment 8. The effects of DNA-binding small molecules destabilizing chromatin began to be studied only 10 years ago. It is not yet clear what structures are formed by DNA-binding small molecules destabilizing chromatin and by what pattern recognition receptors they are recognized to induce type I IFN signaling. Following your advice we have expanded the Introduction and added our hypothesis concerning some associations of the existing effects. We also demonstrate these associations in figure 6 in the Discussion.
Comment 9. Extensive information on how different PSMs activate interferon signaling is included in this section. A brief description of the most important findings at the start or end of this section could make it easier to read and assist readers in identifying the primary conclusions.
Response for the comment 9. Providing the published data on how different PSMs activate interferon signaling we did it very briefly, just pointing out the influence of different PSMs on some pattern recognition receptors involved in type I IFN signaling activation. This field of investigations is at the very beginning and it is impossible to generalize it nowadays as different proteins are considered for different PSMs.
Comment 10. The rationale for the order of PSM presentation, particularly regarding histone depletion, could be introduced earlier in the results section to clarify the sequencing.
Response for the comment 10. We did it at the beginning of the Results:
As the main goal of our study was to compare PSM effects on IFN activation and LINE1 expression with their ability to cause linker histones H1.2 and H1.4 evictions from chromatin described previously [30], PSM order for the effect presentation in all the figures was as follows: 1-8 – PSMs causing intensive linker histones eviction from chromatin (mainly H1.2, but accompanied with H1.4), 9-11 – PSMs causing significant H1.4 eviction from chromatin, but insignificant depletion of chromatin-bound H1.2, and 12-15 – PSMs unable to cause both H1.2 and H1.4 eviction from chromatin.
Comment 11. In Figure 1, please consider a consistent y-axis scale across panels B and C to facilitate comparison between different PSMs.
Response for the comment 11. In Figure 1 we show two different parameters, which are usually analyzed when flow cytometry is used to evaluate the effects. At first, in panel B we demonstrate the change of fluorescent cell proportions (%) and in panel C we demonstrate the change of the mean fluorescent intensity of the cells. We have simplified the Y-axis label in panel B and the legend to make the results more understandable.
Panel B: Proportions of the cells expressing mCherry under the treatment with PSMs. C. Mean fluorescence intensity of mCherry per cell (normalized to control).
New legend to figure 1:
Figure 1. Flow cytometry data for the expression of mCherry driven by IFN sensitive responsive element (ISRE) in HeLa TI ISRE-mCherry cells after PSM treatment for 24 h. A. Color-numeric designation of PSMs and their non-toxic concentrations. Ctr- control; IFN- IFN-α, 103 U/ml; 1- fisetin, 27µM; 2- quercetin, 10µM; 3- resveratrol, 50µM; 4- berberine, 10µM; 5- genistein, 60µM; 6-naringenin, 52µM; 7-delphinidin, 100µM; 8- curcumin, 7.5µM; 9- kaempferol, 2µM; 10- sanguinarine, 0.8µM; 11- EGCG, 65µM; 12- coumarin, 260µM; 13- ginsenoside Rb1, 30µM; 14- thymoquinone, 3µM; 15- apigenin, 5µM. This color-number legend is used in all figures. B. Proportions of the cells expressing mCherry. C. Mean fluorescence intensity of mCherry per cell (normalized to control). The data are presented as an average value ± SD. Significance of the differences between control untreated cells and PSM treated cells was determined using ANOVA test and Dunnett’s post hoc test: significant difference, *— p < 0.05, **— p < 0.01, ***— p < 0.001, ****— p < 0.0001.
Comment 12. The time-course data in Figure 2 provides valuable information. Additional interpretation of the different temporal patterns observed among various PSMs would further enhance the analysis. The y-axis labels are cut off in the figure 2. Please adjust the figure layout to ensure all labels are visible.
Response for the comment 12. We agree that time dependence of the effect observed is an interesting and important information. However, it is not clear yet what peculiar structures are formed by DNA-binding small molecules destabilizing chromatin and by what pattern recognition receptors they are recognized to induce type I IFN signaling. Thus, from our point of view, any interpretation of time dependence will be very speculative and we decided to provide all the data obtained with the interpretation concerning only well demonstrated associations. We demonstrate all of them in the Figure 6 of the Discussion.
In figure 2 we have changed the Y-axis label in panels and the legend.
New legend to figure 2:
Figure 2. Flow cytometry data for the expression of mCherry driven by IFN sensitive responsive element (ISRE) in HeLa TI ISRE-mCherry cells after PSMs treatment for 1, 6 and 24 h. Ctr- control; 1- fisetin, 27µM; 2- quercetin, 10µM; 3- resveratrol, 50µM; 4- berberine, 10µM; 5- genistein, 60µM; 6-naringenin, 52µM; 7-delphinidin, 100µM; 8- curcumin, 7.5µM; 10- sanguinarine, 0.8µM. A. Proportions of the cells expressing mCherry. B. Mean fluorescence intensity of mCherry per cell (normalized to control). The data are presented as m± SD. Significance of the differences between control untreated cells and PSM treated cells was determined using ANOVA test and Dunnett’s post hoc test: significant difference, *— p < 0.05, **— p < 0.01, ***— p < 0.001, ****— p < 0.0001.
Comment 13. The results from the RT-PCR array are presented, yet the biological significance of gene expression changes is not fully explored. The analysis would be improved by classifying the affected genes into functional groups and discussing their consequences.
Response to the comment 13. We provided the classification of the affected genes into functional groups in the Table 1. Analysis of the changes of 84 genes involved in type I IFN signaling for 15 different PSMs. Thus, we would have to consider 84x15 =1260 effects, which is beyond the goals of this study. We generalized the results for every PSM showing under its treatment how many genes increased their expression significantly, how many genes increased their expression by two times and more and, at last, how many genes decreased the expression. However, we provide in the supplements the table with exact results of expression changes caused by every of PSMs for all the 84 genes so that anyone could analyze a single PSM effect on the group of genes of interest.
Comment 14. The authors do not provide enough detail on the statistical analysis methods used. Including this information, along with justifications for chosen tests, is necessary.
Response for the comment 14. We have reorganized the «Statistical Analysis» trying to follow your advice.
We compared the data from the experimental and control groups using one-way analysis of variance (ANOVA) and Dunnett's post hoc test. Differences between groups were considered significant at a p-value <0.05. The basis for statistical processing of results to determine the presence of statistically significant differences between several groups for one independent variable is the randomness of the samples, the equality of the sample size and the normality of the distributions of the samples used. The normality of data distribution was assessed with the Kolmogorov–Smirnov test. Statistical analyses were performed using GraphPad Prism 8.3.0 (GraphPad Software Inc., San Diego, CA, USA).
Comment 15. Certain figures, like Figure 1, may be overly complex. Splitting into multiple figures or using alternative visualization methods might improve interpretation.
Response for the comment 15. In Figure 1 we have changed the Y-axis label in panel B and the legend to make the result interpretations easier. Panel B: Proportions of the cells expressing mCherry in control and treated by compounds of interest distributions of cells. C. Mean fluorescence intensity of mCherry per cell (normalized to control). Moreover, splitting of the figures into multiple figures cannot simplify them, the figures would become even more complex.
Comment 16. Clearly explain the results in light of the body of knowledge already available on PSMs, chromatin remodeling, and interferon signaling.
Response for the comment 16. We did, describing current knowledge of linker histone role in genome functioning, DAMPs and PRR role in IFN signaling activation.
Comment 17. Add recent studies from 2020-2024 to ensure the discussion reflects the latest advancements in the field. Address any limitations of the study and suggest avenues for future research.
Response for the comment 17. We added several recent studies from 2020-2024, which are available.
Comment 18. Ensure all abbreviations are defined upon first use, and consider providing a list of abbreviations for reader convenience.
Response for the comment 18. We added a list of abbreviations for our readers’ convenience.
Comment 19. A thorough proofreading is recommended to improve the clarity and flow of the manuscript.
Response for the comment 19. The revised version has been edited by a professional translator.
Comment 20. The authors link histone depletion, interferon signaling, and LINE1 expression, causal relationships are not definitively established. Additional experiments or a more cautious interpretation would be helpful to improve the quality of the manuscript. Consider performing additional statistical analyses, especially for gene expression data, to further support the conclusions.
Response for the comment 20. We separated 15 PSMs in three groups: (1) causing significant eviction from chromatin of linker histones H1.2; (2) causing depletion of the only chromatin-bound linker histone H1.4 and (3) not inducing eviction of linker histones H1.2 and H1.4 from chromatin. We demonstrated clear difference in the level of type I IFN signaling activation between these groups. It seems not appropriate as correlation between two effects of the compound cannot be analyzed when only two doses are used. Now we have revealed the effects comparing tree different groups of PSMs and in future we could devote special studies for analysis of single PSM effects by correlation analysis, although it cannot show causative relations of the effects.
Once again thank you for consideration of our manuscript.
Sincerely yours,
Olga Vlasova, MSc, junior scientist
of the Department of chemical carcinogenesis,
Institute of Carcinogenesis
Blokhin National Medical Research Center of Oncology
+79256761167
Marianna Yakubovskaya, MD, PhD, DSc,
Head of the Department of chemical carcinogenesis,
Institute of Carcinogenesis
Blokhin National Medical Research Center of Oncology
+79256761167

Round 2
Reviewer 1 Report
Comments and Suggestions for Authors
International Journal of Molecular Sciences (Manuscript ID: ijms-3293243), Comments to the Authors:
Title: Anticancer Plant Secondary Metabolites Modulating Chromatin Activate Interferon Signaling and LINE1 Expression
Comments
After reading the authors response to my comments, I think the revised paper can be accepted for publication.
Author Response
To: Peer Reviewer, 03.12.2024
Dear Peer Reviewer,
We are very grateful for your careful reading of our manuscript and our answers, and as well as for appreciation of our study.
Once again, thank you very much!
Sincerely yours,
Olga Vlasova, MSc, junior scientist
of the Department of chemical carcinogenesis,
Institute of Carcinogenesis
Blokhin National Medical Research Center of Oncology
+79256761167
Marianna Yakubovskaya, MD, PhD, DSc,
Head of the Department of chemical carcinogenesis,
Institute of Carcinogenesis
Blokhin National Medical Research Center of Oncology
+79256761167